# Clients' and genetic counselors' perceptions of empathy in Japan: A pilot study of simulated consultations of genetic counseling

Chikako Tomozawa[1]*, Mikiko Kaneko[2], Motoko Sasaki[1,3], Hidehiko Miyake[1,3]

1 Division of Life Sciences, Department of Genetic Counseling, Graduate School of Humanities and Sciences, Ochanomizu University, Bunkyo-ku, Tokyo, Japan, 2 Department of Clinical Genetics, The Jikei University Hospital, Minato-ku, Tokyo, Japan, 3 Genetics Division, Institute for Human Life Science, Ochanomizu University, Bunkyo-ku, Tokyo, Japan

* chikabom0108@gmail.com

**Data Availability Statement:** The data that support the findings of this study are available on request from the Institutional Review Board of the Humanities and Social Science Research Ethics

## Abstract

The rapidly increasing availability of genetic testing is driving the acceleration of genetic counseling implementation. Empathy is important in medical encounters in general and forms a core component of a successful genetic counseling session; however, empirical evidence on empathy in genetic counseling is minimal. This study aimed to explore the perceptions of empathy in simulated genetic counseling consultations from the perspectives of clients and genetic counselors. Semi-structured interviews and interpersonal process recall were used with participants of simulated genetic counseling consultations to elicit their experiences of empathy. A constructivist grounded theory was used for data analysis. A total of 15 participants, including 10 clients and 5 genetic counselors, participated in 10 simulated counseling sessions. The genetic counselors attempted to demonstrate empathy and were sensitive toward detecting changes in clients. Meanwhile, the clients' perceptions represented their feelings and thoughts elicited through the counselors' empathic approaches. This was the first process study to examine empathy in simulated genetic counseling sessions. Our model of communication of empathy is a process in which counselors try to address implicit aspects of clients, and clients are provided with time and a safe place for introspection, which contributes to discussions on building good relationships with patients. There is also a suggestion of the utility of simulated consultations for healthcare providers to learn empathic communication.

## Introduction

Empathy is a significant keyword in patient-centered practice. In the medical context, empathy and communication as well as their benefits have been the subject of much discussion [1–4]. While empathy is emphasized as a professional skill, the construct of empathy is discussed in many academic fields because of its ambiguity. Specifically, it depends on the experience of the recipient or the provider-patient relationship, as evidenced by empirical studies involving

Committee in Ochanomizu University (e-mail: kenkyo-TL@cc.ocha.ac.jp) and the corresponding author. The data are not publicly available due to privacy and ethical restrictions. Parts of data that support the findings of this study are available in this article.

**Funding:** The authors received no specific funding for this work.

**Competing interests:** The authors have declared that no competing interests exist.

various professions, such as studies involving psychotherapists [5, 6], teachers [7, 8], physicians [1, 9], and nurses [10, 11].

Different features of recipients' perspectives on empathy are reported in different patient-provider relationships. Bachelor [5] reported that patients perceived their psychotherapist's empathy as cognitive (i.e., the psychotherapist recognizes the patient's experience, state, or motivation), affective (i.e., the psychotherapist feels what the patient is feeling), sharing (i.e., the psychotherapist discloses personal opinions or experiences relevant to the patient's communication), and nurturant (i.e., the psychotherapist's supportive, security-providing, or totally attentive presence). The grounded theory analysis by MacFarlane et al. [12] yielded a model consisting of three clusters: (1) relational context of empathy, (2) types of empathy, and (3) utility of empathy. Regarding empathy in the physician-patient relationship, the patient's view of the physician's empathy is linked with the patient's satisfaction with their physician, interpersonal trust, and adherence to the physician's recommendations [13]. Another study revealed that patients with cancer classify empathic clinicians' behaviors into five categories: relationship sensitivity, focusing on the whole person, communication, clinician attributes, and institutional resources and care processes [14].

Genetic counseling is a professional practice that is rapidly growing with emerging genetic technologies. The profession is defined by a therapeutic relationship with an empathic understanding of the client/patient's concerns and needs [15]. In theoretical literature for genetic counseling students, empathy is identified as one of the most important techniques in psychosocial counseling [16–18]. As per the existing genetic counseling theory, empathy plays a vital role in the establishment of counselor-patient relationships and can be helpful because it can offer more direct interpretations of patients' inner experiences [19, 20]. It has also been empirically shown that Japanese certified genetic counselors consistently attempted to utilize listening and reflection skills for empathy [21]. However, clients' perspectives on their empathy experiences have not been addressed in the context of genetic counseling. There are also few practical tools for learning/developing empathy-related skills [22].

Although some published studies have suggested the significance of empathy in healthcare providers, including genetic counseling professions, there is a lack of empirical data about empathy in genetic counselors. The aim of this pilot study was to explore the process through which counselors and clients experienced empathy in the genetic counseling context by analyzing simulated consultations. Our research questions were as follows: How do clients perceive empathy by genetic counselors? How do genetic counselors practice empathic understanding in a simulated consultation? How do these two perspectives relate to each other? We conducted a qualitative study and adopted the interpersonal process recall (IPR) technique to clarify the sequential process of empathy, previously unidentified in the genetic counseling context.

## Materials and methods

### Study design

This study adopted IPR interviewing, which uses video-assisted recall to access conscious yet unspoken experiences in professional-client interactions [23]. The interviewer and the participant look at what is happening immediately before and after the actual event that is under review. IPR is considered a powerful approach for accessing a client's thoughts and feelings in psychotherapeutic counseling [24–26]. This method can be usefully applied to process studies of genetic counseling [27–29]. Due to the difficulties associated with using IPR for clients in clinical settings, our survey targeted simulated genetic counseling consultations.

## Researchers

The first author, a doctoral student in genetic counseling conducted all the interviews. The research team consisted of cisgender individuals: the first author (female), the first author's research advisor (male), and two genetic counselors (females) with over ten years of clinical experience. The first author and two female co-researchers have been certified Japanese genetic counselors, who have been educated on the clinical use of empathy through the training program; thus, their clinical practice was mostly informed by a client-centered approach. The first author's research advisor has a Japanese certification as a clinical geneticist.

## Participants

First, we divide the client scenarios into three categories: prenatal testing, developmental disability, and hereditary cancer, which are listed in Table 1. We assumed that the majority of candidates would be female students, due to the affiliation of the first author to a women's university and the recruiting process being commenced by her acquaintances. Thus, the client scenarios were set in a female-centric environment to allow for the smooth recruitment of participants. To avoid any psychological burden on the participants, they were required to not have a history of mental illness-related medical examinations, or sleep problems that were defined by difficulty falling asleep or staying asleep and waking up too early over the last two weeks. The inclusion criterion for genetic counselors was having clinical experience of at least five years in genetic counseling. A total of 10 simulated consultations were conducted with 15 participants (10 participants in client roles and five in counselor roles; all participants were females as biological sex). Five out of 10 participants in client roles, and four out of five participants in counselor roles were recruited through the acquaintances of our research team members. The remaining participants were selected by snowball sampling to participate in the study. Participants both in client roles and counselor roles were informed that the study would focus on empathy in genetic counseling sessions, and that the sessions they participated in would be recorded for interviews. After participants were provided the above-mentioned information about the study, they agreed to participate.

**Table 1. Client scenarios of simulated genetic counseling consultations.**

**Setting A: Prenatal testing**

The client is currently 11 weeks pregnant. She wants to undergo prenatal testing. When she discussed this with her physician, he informed her that genetic counseling is available in the hospital. She made an appointment for genetic counseling to obtain a detailed explanation about prenatal testing (non-invasive prenatal genetic testing, Quattro testing, combined testing, amniotic fluid testing, and trophoblastic testing can be performed in the hospital).

**Setting B: Developmental disability**

The client is currently dating on the premise of marriage. She has learned that one of her partner's siblings has a developmental disability and so, she has made an appointment for genetic counseling to discuss whether or not the developmental disability is inherited.

**Setting C: Hereditary cancer**

The client's mother was diagnosed with right breast cancer two years ago. A few months ago, she was diagnosed with a recurrence. At the same time, her doctor indicated that there were many patients with cancer in her family history. She decided to pursue genetic testing to clarify the possibility of hereditary breast cancer, and a pathogenic variant was detected. The client made an appointment for genetic counseling at the hospital (her mother's hospital) to discuss whether she should pursue genetic testing.

Note: Three detailed background scenarios were provided for scenarios A to C. Participants in the client role were asked to select the scenario that they felt most familiar with.

## Simulated genetic counseling consultations

A simulated consultation was performed by one client and one counselor. The participants acting as clients were provided with the client scenarios (Table 1). They were asked to choose the scenario which they felt a sense of familiarity toward, to make it easier for them to participate in the simulated consultations. They were also warned that choosing a scenario that was too familiar could result in possible mental stress. The participants acting as counselors were informed of the scenario chosen by the participants acting as clients and were paired with them at least one week in advance. Each simulated consultation began with the designed dialogue of a client role. All sessions were conducted online via Zoom and recorded using the program's recording function.

## IPR interviews

The author conducted the IPR interviews online with each interviewee as a genetic counselor or a client individually in a quiet, comfortable, and private room. The IPR interviews took place within 48 hours of the video-recorded conversation to activate the memories more vividly and easily [23]. During the explanation prior to study participation, we presented a definition of empathy wherein one puts oneself in the other person's shoes, understands how he/she is feeling and thinking, and shares that understanding. The interviews were conducted using the interview outline (Table 2) that followed the existing literature [12]. This outline served as a guide, yet the interviewer remained open and flexible to bring out the interviewee's responses in more detail [30]. During IPR, the interviewer attempted to have the interviewee identify and focus on specific moments regarding empathy by themselves during the video-recorded interaction to elicit interviewees' thoughts and feelings regarding the moment [23]. The interviewer encouraged discussions regarding each area of inquiry with each respondent, but also pursued particular areas that arose for each interviewee in more depth [30, 31]. The interviewer thus facilitated this process by showing the sections that included topics regarding the clients' feelings or counselee/counselor's non-verbal cues, as necessary. For example, "What did it mean to you when the counselor said, 'I'm sure you must have been surprised'?" (When

**Table 2. Outline of the IPR interview.**

| **1. Introduction** |
| --- |
| • This study focuses on your empathic experiences in genetic counseling. |
| • We hope you will help us better understand clients' inner experiences regarding empathy. |
| **2. Conceptual framework** |
| • What does empathy mean to you? |
| • Prompt (as necessary): Empathy can mean that you feel what another person feels. |
| In the context of genetic counseling, empathy may mean that your counselors put themselves in your shoes and attempt to understand you really well, and that you feel that they get you. |
| **3. Session Review** |
| • For clients: Did you, during this last session, experience that your counselor was really getting you or really understanding you (use respondent's words when possible)? |
| • For counselors: Tell me, during this last session, a scene/situation where you attempted to understand the client or show your empathic understanding toward the client. |
| • Discussion |
| • Video review |
| • When empathy is identified, stop the video for discussion |
| • Primary areas of inquiry: Momentary feelings and thoughts regarding empathic experiences: What did you feel from the other person's words or gestures? |

a counselor made a comment related to the client's feelings); "In this section, you [client] appeared to be thinking, how did you feel at this time?" (When a client stopped talking and looked upwards); or "What was the purpose of your question at this time?" (When a counselor asked a client about their family history and background of a consultation). The interviews were recorded using a digital recorder and then transcribed verbatim for data analysis by the interviewer as soon as each interview was finished.

## Ethics approval

This study was approved (No. 2020–137) by the Review Board of the Humanities and Social Science Research Ethics Committee of the institution to which the first author is affiliated. Participants were briefed on the content of the interview at least two days prior to the interview. Participants gave their informed consent by e-mail prior to participating in the interview.

## Data analysis

Constructivist grounded theory [32] was used to analyze the data collected through the IPR interviews. This approach was chosen for two reasons. First, grounded theory requires researchers to use theoretical sampling procedures whereby the data they initially collect are employed to inform subsequent data collection and analysis. This procedure ensures consistency in coding. Second, grounded theory allows researchers to produce a theory of a process. This approach was thus appropriate for our study, which aimed to generate a new theory for the process of empathy. In this study, the existing theory of empathy in the genetic counseling context shows that empathy plays a vital role in establishing patient-counselor relationships and can be a helpful tool for interpreting patients' inner experiences [19, 20].

The use of grounded theory analysis entails a set of methods to enhance trustworthiness. We adopted some steps to maximize trustworthiness, including utilizing co-researchers, the constant comparative method [33], and abductive steps including additional interviews [32]. Details of these steps are as follows: We performed inductive analysis during data collection. Our coding procedure included the following two steps: initial coding and focused coding. Incident-by-incident coding was used to conceptualize initial codes. Next, the first author used focused coding by grouping or selecting significant identified codes from initial codes and built categories using selective coding. Throughout the analysis, we applied the constant comparative method [33] and memo-writing [32]. Lastly, a theory that interpreted the relationships between the categories and codes was generated [32]. Simultaneously, we added abductive steps, asked about possible new aspects through additional interviews to test the hypothesis of our data. Coding and theorization were conducted by the first author. The other authors checked the transcriptions, analyzed the data, and pointed out any codes that needed to be corrected. Research team members discussed the first author's interpretations and theorization to enhance credibility and confirmability. The first author's coding and theorizing procedures and team discussions were repeated to reach an accurate interpretation of the data. The constant comparative method was used in an iterative process each time more data became available following another interview. We adopted the concept of theoretical sufficiency, which seeks the point at which the researcher has sufficient depth of understanding to address the study aims [32]. In the context of this study, theoretical sufficiency implied the point at which various participants' experiences that encapsulated the process of empathy in simulated genetic counseling consultations were captured. Data collection was completed upon theoretical sufficiency, that is, when sufficient richness of information was achieved to address the study aim. Since the data in this study were in Japanese, all the quotes used in this article were translated by the first author and back-translated by a professional translator,

Editage (www.editage.com). Thereafter, we checked the back-translated quotes to determine whether the nuances of the participants' experiences were reflected in the English translation.

## Results

### Overview of participants and simulated genetic counseling consultations

A total of 10 simulated consultations were conducted (Table 3). The duration of the client interviews ranged from 28 to 47 minutes (median = 34), and the duration of the counselor interviews ranged from 55 to 70 minutes (median = 58).

### Clients' experiences of simulated genetic counseling consultations

The first research question inquired how clients perceive empathy by genetic counselors. Five theoretical categories were generated: *having concerns about a given theme*; *having concerns about a first experience*; *gaining awareness and thinking due to the counselor's approach*; *mind relaxing due to counselor's questions and responses*; and *establishing a good foundation and having a better outlook* (Table 4).

**Having concerns about a given theme.** Clients arrived at the simulated consultations with their own images, feelings (e.g., "being anxious," "being so nervous"), thoughts (e.g., "being confused," "try to listen to the counselor"), and stances about client roles (e.g., "being passive"). In other words, they participated in the consultations after some preparation for their pre-selected scenarios. This category involved the following representative codes: *coming to the session with a positive attitude; I guess we are a cancer family*; and *concerned about differences in perception with her partner*.

**Table 3. Overview of the participants and simulated genetic counseling consultations.**

| Session No. | Clients | | Genetic counselors | Simulated settings |
|:---:|:---:|:---:|:---:|:---:|
| | **Attribute** | **Knowledge of genetic counseling** | **Years of clinical experience** | |
| 1 | Master's student, 23 years | Knew | Over ten years | Hereditary cancer |
| | | (with no experience in genetic counseling) | | |
| 2 | Master's student, 22 years | Knew | | Hereditary cancer |
| | | (with no experience in genetic counseling) | | |
| 3 | Master's student, 22 years | Knew | Five to ten years | Developmental Disability |
| | | (with no experience in genetic counseling) | | |
| 4 | Master's student, 22 years | Knew | | Developmental Disability |
| | | (with no experience in genetic counseling) | | |
| 5 | Master's student, 22 years | Knew | Five to ten years | Hereditary cancer |
| | | (with no experience in genetic counseling) | | |
| 6 | Master's student, 24 years | Knew | | Prenatal testing |
| | | (with no experience in genetic counseling) | | |
| 7 | Master's student, 24 years | Knew | Over ten years | Developmental Disability |
| | | (with no experience in genetic counseling) | | |
| 8 | Master's student, 24 years | Never knew | | Developmental Disability |
| | | (with no experience in genetic counseling) | | |
| 9 | Master's student, 23 years | Have heard | Over ten years | Prenatal testing |
| | | (with no experience in genetic counseling) | | |
| 10 | Master's student, 24 years | Never knew | | Hereditary cancer |
| | | (with no experience in genetic counseling) | | |

**Table 4. Categories, representative codes, and quotes extracted from clients' and genetic counselors' responses.**

| Interviewee | Theoretical categories | Representative codes | Representative quotes |
|---|---|---|---|
| **Clients** | Having concerns about a given theme | Coming to the session with a positive attitude | In my mind. . .it did not seem too negative. It was a situation where this client would have thought, "Let's just listen anyway." (Session No. 9) |
| | | I guess we are a cancer family | Before I started counseling, I had thought that I was from a cancer family. However, I did not think that I had to deal with it. I thought that I would probably get cancer when I got older, and I was worried about it, but I thought that if it was in my family, it could not be helped. (Session No. 10) |
| | | Concerned about differences in perception with her partner | My biggest fear is that if the fetus has a disease, my partner might tell me to have an abortion. When my partner asked me to get tested, I was very afraid of getting caught up in that possibility. This time it is a survey situation, but I think I would feel the same way if I were pregnant. (Session No. 5) |
| | Having concerns about a first experience | Feeling anxious before the first experience | I was very nervous because I imagined that the client was very anxious, being new to genetic counseling, pregnancy, and everything else. I remember feeling very anxious about what to say first. (Session No. 9) |
| | | Do not understand the role of a genetic counselor | I was wondering what kind of position a genetic counselor is. To me, doctors have the image of being very knowledgeable about diseases. As we talked this time, I wondered whether they provide knowledge about the disease, or mainly provide counseling. (Session No. 2) |
| | | Do not know what information would be provided | I was not in a simulated setting where I was anxious about developmental disabilities. So, I had a simple question about what developmental disabilities were, and what would she say to me about it? (Session No. 8) |
| | Gaining awareness and thinking due to the counselor's approach | Client's inner thoughts revealed by the counselor's words | The counselor asked me questions such as, "Do you have any idea about the background of this?" Then I had the feeling that something I had inside of me was gradually becoming clearer. (Session No. 7) |
| | | Information provided by a counselor led clients to think for themselves | At first, I was only concerned about my mother, who has cancer. But when the counselor told me that "you and your family members also are at risk for cancer," I came to myself and realized that I had come here today to talk about myself. That calmed me down a bit and I prepared to face myself. (Session No. 10) |
| | Mind relaxing due to counselor's questions and responses | Becoming easy to talk after being asked a question | When she summarized what I said, she also asked me if I had any concerns. She had given me a chance to say it, so, I felt I had received a message from her that she was willing to listen to me. When she asked me if I had any concerns, it was easy for me to naturally express what I was worried about. (Session No. 4) |
| | | Nervousness being reduced by counselor's listening | While I was talking for a long time, she listened to me while nodding her head, which made me feel that it was okay to talk about what I usually think. I think the fact that she listened to me made me feel at ease. (Session No. 5) |
| | | Reassurance of being understood | I felt very relieved when the counselor verbalized my purpose for coming, in plain language, "You came here to find out these things." She understood. (Session No. 3) |
| | | Anxiety caused by a typical brief silence in online counseling | I think that the counselor got my message. However, I still felt a little uneasy when brief pauses sometimes came in the middle of talking. I have been a little anxious, so when the conversation stopped, I wondered if what I said was strange. Maybe that's the online influence. I had those anxieties, but not often, only very occasionally. There was nothing else to worry about. Only the brief silence bothered me a little. (Session No. 8) |

(*Continued*)

**Table 4.** (Continued)

| Interviewee | Theoretical categories | Representative codes | Representative quotes |
|---|---|---|---|
| | Establishing a good foundation and having a better outlook | Making the decision to get genetic testing after getting cancer risk information | I knew that most people with breast cancer can survive after early detection and treatment, so I did not feel uncomfortable getting a genetic test. But when I heard from the genetic counselor that 70% to 80% of people with a genetic mutation developed breast cancer and the risk compared with the general population, I made up my mind. I thought I should get the genetic testing. (Session No. 4) |
| | | Clarifying future direction | After listening to the counselor, I felt that I did not need to be so anxious. It made me think that there is no need to tell my partner now. I think I have a clearer picture of how I am going to resolve this issue with my parents. (Session No. 8) |
| | | Still having some concerns | I was very satisfied regarding the information on breast cancer, but I still had a few concerns about ovarian cancer. If I got a positive result, I would like to get surveillance periodically and try hard to ensure early detection of cancer. But the counselor told me that it is difficult to detect ovarian cancer in the early stage, that made me slightly nervous. (Session No. 2) |
| **Genetic counselors** | Making use of one's experience as a genetic counselor | Having their own thoughts about empathy and empathic understanding | I have always thought that empathy is different from empathic understanding. In genetic counseling, I try to communicate empathic understanding. Empathy, I wonder what it is. First of all, I talk on the premise that I don't know how the other person really feels. But as we talk, I can understand in my own way why the clients felt that way in their situation. I understand that this is empathic understanding. (Session No. 1) |
| | | Talking while feeling that I am nervous in a counseling session outside of my specialty area | Since I don't have much experience with patients with breast cancer and cancer genetic counseling, I was a little more nervous than usual, wondering what kind of questions I would be asked. (Session No. 6) |
| | | Understanding the client's behavioral characteristics by her attitude during my explanation | I thought she must not be the type of person who nods often, but only when she understands sufficiently. (Session No. 4) |
| | Facilitating counseling that is sensitive to a client's mood | Facilitating the counseling while being careful about the client's understanding | I thought that she did not talk a lot compared to the first one. I also understood that she gave a matter-of-fact answer. Therefore, I was worried that I'm not sure if I can ascertain her level of understanding of my explanation. (Session No. 6) |
| | | Feeling smooth communication by noticing a slight difference in a client's responses | I felt it was a little different compared to the first half of the session, for example, her phrasing. I think I had been talking while feeling that she was listening to me. There might not be many words, but I felt like we were playing catch. I felt something like that. (Session No. 2) |
| | Imagining a client's background | Trying to understand clients' background for empathy | The first step in the introduction is to hear why the client came to this consultation. It is sometimes difficult to empathize with someone if you do not know their background. (Session No. 7) |
| | | Wanting to know a client's thoughts about cancer | She was a healthy relative of a patient with cancer, so I wanted to hear her thoughts or fears about heredity. The point that I wanted to hear most about was her image of cancer. I thought that her image of breast cancer would affect how she received the test results. (Session No. 10) |
| | Adjusting oneself to fit a client and approaching them | Recognizing that my choice of words and manner of providing information are affected by the client's behaviors | I have essential information to tell a client, you know? When I provide it, I am affected by the client's thoughts or the client's perspective. Maybe, I choose my own, for example, how to tell, choice of words. (Session No. 2) |
| | | Wanting to check if I correctly understood what a client was saying | I worried about whether I understood what a client was saying. I proceeded while checking the situation. (Session No. 5) |
| | | Understanding from her words that she understands an explanation and is ready to accept her test result | At the beginning, she said that she must have gotten cancer because of her genetic background and that she was a little scared. When she said, "If it means that I can take preventive measures," I felt that she understood my explanation and that if she was going to get tested, she would be able to accept the results properly. (Session No. 10) |

*(Continued)*

**Table 4.** (Continued)

| Interviewee | Theoretical categories | Representative codes | Representative quotes |
|---|---|---|---|
| | Being sensitive to a client's changes | Noticing that a client's words do not match those at the beginning | I was surprised that she had a negative attitude toward having children, because she initially said she would have a baby whatever the test results might be. I felt that there might be something different from her initial words then. (Session No. 5) |
| | | There was no change suggesting that our relationship deepened | There was no change in her attitude until the end. When you see a slight change, or a change in the client's viewpoint, there is something you can sense, but this time there was no change. The client had a positive attitude at the beginning, and I don't think she ever expressed negative feelings, but I didn't get any sense that our relationship had deepened. (Session No. 1) |
| | | I still cannot establish a client's true concern | The client's partner had a sibling with a developmental disability, and her partner did not have a disease, but both the client and her parents were concerned about the disease. It was not clear to me whether they were worried that her partner might also develop the disease in the future, or whether they were only worried about her children in the future, in the next generation. I did not understand that point. (Session No. 8) |
| | Picturing future direction from a client's perspective | Managing the session by imagining the client's future action | I was sure that she would explain this to her family. When I imagined that, I wanted her to take something home for her explanation. A one-time session may not be sufficient. I thought it would be better if it would lead to the next one. (Session No. 8) |

**Having concerns about a first experience.** At the beginning of the dialogue with the counselor, the client had her own images and "unsureness" about the first meeting with a new person at the first genetic counseling session. Negative feelings might be magnified, as indicated by the participants' recall of feelings of nervousness and anxiety: "I only had a negative feeling. I could not be positive." They were also concerned about how they should behave because of their uncertainty regarding their first time in the situation. The "unsureness" could be seen in the following codes: *feeling anxious before the first experience*; *do not understand the role of a genetic counselor*; and *do not know what information would be provided*.

**Gaining awareness and thinking due to the counselor's approach.** As the dialogue continued, the client received the "counselors' techniques," of which details were shown through the theoretical category, "*adjusting oneself to fit a client and approaching them*." The clients received new information and sometimes recognized the difference between that information and their own understanding: "She (counselor) explained cancer risk information and said, 'you may have an additional risk of ovarian cancer.' I realized that I did not understand until she pointed out." Especially in the process of hearing their background and family history, the clients began to think for themselves. For instance, a client recalled that "I was passive at the beginning of the consultation, but I was able to think about the cancer risk for myself by talking about my family history." This process, which included repeatedly being asked questions, answering them, and having their statements confirmed by genetic counselors, enabled the clients to view their situations objectively. Questions from another person's viewpoint enabled clients to think in a way that they had never considered on their own: "Her question enabled something hidden inside of me to be clear." These aspects were also shown in the following representative codes: *clients' inner thoughts revealed by the counselor's words* and *information provided by a counselor led clients to think for themselves*.

**Mind relaxing due to counselor's questions and responses.** In parallel with "gaining awareness and thinking," clients experienced peace of mind. Not only the counselor's questioning but also the verbal exchange itself caused clients to relax: "She rephrased my purpose

of the consultation in simple words. I was relieved. I thought that she understood me." The counselor's familiarity, acceptance, and commitment toward understanding clients' experiences were perceived by the client. Repetition of these exchanges transformed nervousness in the early stage of a session into reassurance, safety, or a good feeling about the counselor, and concerns about how to articulate thoughts were relieved: "I felt that I could tell her anything." These changes could also cause the expression of questions that came to their minds during the session: "I was able to ask any questions." This could be described as a change from an "unsure" state to one of relaxation. This category included an experience of feeling that the counselor was by their side and feeling understood and accepted by the counselor: *becoming easy to talk after being asked a question*; *nervousness being reduced due to counselors' listening*; and *reassurance of being understood*.

As a session progressed, clients received information that could reduce their anxiety, or help them recall their thoughts and feelings: "I feel somewhat relieved when she told me the frequency in the general population. It was nobody's fault"; "Her explanation for follow-up guidance was very helpful because I felt uneasy when I did not know what to do." During this time, the experiences described in the categories above were repeated. Clients sometimes perceived a difference from what they had imagined before the session or felt a sense of discomfort brought about by the online communication. Clients also experienced a return of the anxiety they felt at the beginning of the session. However, the simultaneous experience of their "*mind relaxing*" reduced these negative feelings. The code "*anxiety caused by a typical brief silence in online counseling*" shows this emotional complexity.

**Establishing a good foundation and having a better outlook.** Clients had *concerns about a given theme* at the beginning of a session. As described in the above categories, they experienced changes in the aspects of both psychology (relaxing and feeling understood) and information (increased knowledge and understanding). This made clients' thoughts and feelings organized and made their future image clearer. This was also an experience that allowed clients to have confidence in their own thinking, as represented by the codes: *making the decision to get genetic testing after getting cancer risk information* and *clarifying future direction*. However, there were also unexpressed anxieties and unspoken thoughts in a session: "My question was not answered in the session after all. I thought it could not be helped." After the session, clients sometimes recalled what they had to ask: "Thinking about it now, I was not told the cons of genetic testing." There were narratives of some remaining concerns: "I was very satisfied regarding the information on breast cancer, but I still had a few concerns about ovarian cancer." This indicated that the clients' concerns were not always solved completely.

## Genetic counselors' experience of simulated genetic counseling consultations

The second research question inquired how genetic counselors practice empathic understanding in a simulated consultation. Six theoretical categories were generated: *making use of one's experience as a genetic counselor*; *facilitating counseling that is sensitive to a client's mood*; *imagining a client's background*; *adjusting oneself to fit a client and approaching them*; *being sensitive to the client's changes*; and *picturing future direction from a client's perspective* (Table 4).

**Making use of one's experience as a genetic counselor.** Genetic counselors approached counseling sessions based upon their previous clinical experience, and they had their own thoughts about empathic understanding: "I think that I usually practice empathy using my five senses." When they started a simulated session, they sometimes became aware of discomfort or difficulty due to differences from the norm: "I considered options of genetic testing that were eligible for clients in their twenties. I think that I felt discomfort because the client was

younger than usual." Particularly at the beginning of the session, they tried to identify the client's characteristics and imagined what the client was thinking, based on their experience.

**Facilitating counseling that is sensitive to a client's mood.**   Genetic counselors focused on building good relationships with clients: "I asked questions sensitively and tried to understand the client's situation, which might build a rapport between her and me." They were also continuously thinking about how to provide true support to their clients: "I thought that there must be a good reason that the client came to the counseling session alone. I thought that I should understand her real concern in order to make the counseling fit her needs." In other words, genetic counselors had both a present focus on smooth communication and a long-term perspective on supporting clients. With these perspectives, they *facilitated counseling while being careful about the client's understanding*. They had concerns in their mind about whether their attitude and words made the client feel bad and sometimes felt *smooth communication by noticing a slight difference in a client's responses*.

**Imagining a client's background.**   Genetic counselors imagined a path to decision-making based on their clients' background information: "I asked details about the condition of the client's mother, because I thought that it was significant for her decision to undergo genetic testing in the session." While listening, genetic counselors considered what information should be provided additionally and what aspects of their clients' situations were not being disclosed: "I thought she was nervous about being pregnant or having a baby rather than the genetic testing itself. I thought that her relationship with her husband might be the key to her decision-making." As the dialogue progressed and the clients' information was acquired, genetic counselors could deepen their understanding of the clients.

**Adjusting oneself to fit a client and approaching them.**   Based on the process described above, genetic counselors considered the next steps in information provision, chose their words, and talked to the client. After these actions, they assessed if the client understood the information, checked what they should provide, and adjusted their way of counseling flexibly according to the situation. During this process, genetic counselors might identify an important point that they should inform the client about: "I realized that what she wanted to know could not be checked by genetic testing. I thought that I should tell her that." Some counselors were not sure about how much information they should provide, or how to provide it while being considerate of the client's family relationships: "I was not sure about talking about a specific diagnosis in a situation where the partner's brother with the disease was not included in the session. This was because I did not know the correct name of his diagnosis. Mentioning a specific term might cause misunderstanding." Genetic counselors balanced their various thoughts as the session progressed.

**Being sensitive to the client's changes.**   Genetic counselors noticed that their initial understanding of the client changed. Since "change" was a phenomenon that involved the passing of time, a change in the understanding of clients could be recognized only after a certain amount of communication: "(When the client started to talk in the second half of the session,) I thought that she was willing to share her thoughts or experiences. I felt that our relationships became more intimate." Counselors always observed the client's facial expressions, tone of voice, and so on, but might not always perceive any changes: "It was her facial expression. I could catch the change over the screen"; "I never realized any change in the client during the entire session." Alternatively, they might perceive changes and then recognize that some aspects of the client were still hidden.

**Picturing the future direction from a client's perspective.**   By repeating the processes mentioned above, genetic counselors could consider a client's perspective with a deeper understanding: "I realized that she might be worried about it. I thought whether I should take this

on. I thought about how to manage the session and what kinds of information to provide." Based on this perspective, they *managed the session while imagining the client's future action.*

## The process through which empathy is experienced in simulated genetic counseling consultations

The third research question assessed how clients' and counselors' perspectives relate to each other. Associations between theoretical categories are illustrated in Fig 1, which were confirmed through the questions in the IPR interview for theoretical sampling. Genetic counselors managed the simulated consultations, making use of their experiences and being sensitive to their clients' mood. Four theoretical categories generated regarding genetic counselors' perspectives on demonstrating empathy: *imagining a client's background, adjusting oneself to fit a client and approaching them, being sensitive to the client's changes*, and *picturing future direction standing from a client's perspective*. The counselors' practice contributed to the abovementioned theoretical categories and changes in the clients' perspectives. Clients' experiences of perceived empathy were classified into two major theoretical categories: *gaining awareness and thinking due to the counselor's approach* and *mind relaxing due to counselor's questions and responses*. The process behind clients' perspectives was not one-way, but rather interactive.

## Discussion

There is a lack of empirical research on conveying empathy in genetic counseling. This study reported the experiences of clients and genetic counselors related to empathy in simulated genetic counseling consultations (Table 4 and Fig 1). The categories in this study (i.e., "*gaining awareness and thinking due to the counselor's approach*" and "*mind relaxing due to counselor's questions and responses*") were similar to previous reports of perceived empathy [5, 6, 12]. This

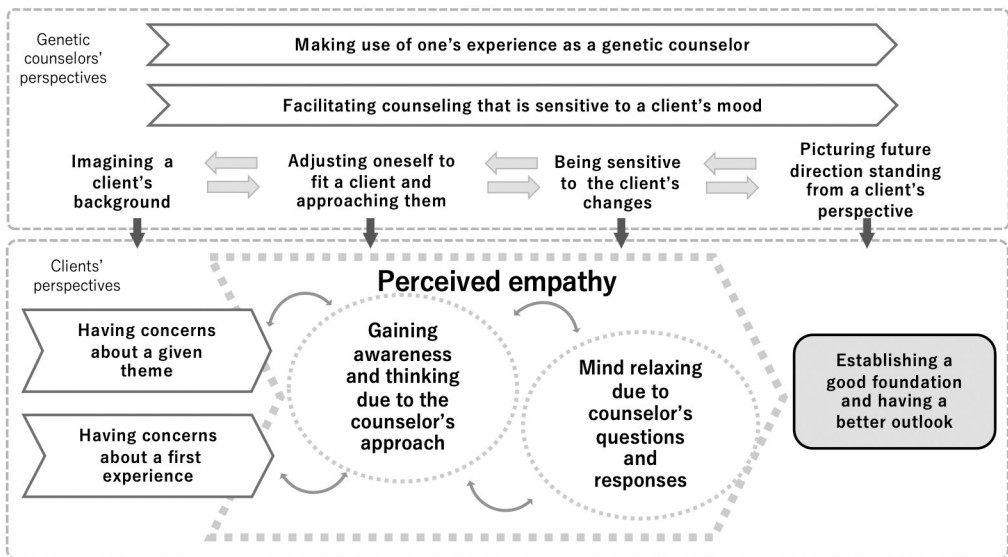

**Fig 1. Clients' and genetic counselors' experiences regarding empathy in simulated genetic counseling consultations.** Categories related to clients' perceived empathy were included in a simulated genetic counseling consultation experience. Bidirectional arrows show that clients' feelings and thoughts alternate between "*having concerns*" and "*having a better outlook*".

similarity suggests the perception of empathy in a simulated consultation could be consistent with real clients' perceptions.

The result that clients' perspectives included "gaining awareness" showed that clients could realize a deeper self-understanding. A concept of self-understanding has also been reported as a form of perceived empathy in previous studies of psychotherapeutic contexts [5, 12]. This shows that the concepts of perceived empathy include a common perception despite having a different communication process in different caring relationships. Clients arrived at this self-understanding by answering counselors' questions. In genetic counselors' perspectives, they approached a client by imagining their background and/or adjusting oneself to fit their situation, which included their intention of showing empathy. Our study showed that genetic counselors' approaches could be useful for clients' self-understanding.

Our codes of "*nervousness being reduced by counselors' listening*" or "*reassurance of being understood*" were in line with findings that clients' feelings of safety and trust contribute to an experience of being empathically understood by psychotherapists [5, 12]. The participants in this study had "*concerns about a given theme*" and "*concerns about a first experience.*" Those concerns could be promoted by the fact that participants in client roles had never experienced a simulation of a counseling session and never met the participants acting as counselors. Although there was a sense of uncertainty at first, the participants acting as clients gradually felt safer and were able to trust genetic counselors, which might have emphasized their perception, consistent with Bachelor's [5] *nurturant* empathy. In the medical communication context, in which patients often attend a first consultation with concerns about their health issues, their experiences of perceiving genetic counselors' empathy may be similar to our results.

The counselors' perspectives revealed their management of the counseling sessions and their sensitivity to the clients' changes. Veach et al. [20] described the skill of advanced empathy as communicating an understanding of implicit aspects of patient experience. This study, targeting simulated clients and counselors, demonstrated how implicit aspects are addressed and recognized from both perspectives, which suggests that simulated consultations might be useful tools to learn how to communicate empathic understanding or evaluate learners' skills around empathy. Jacobs and McEwen also discussed the utility of simulation for learners to develop effective client-counselor relationships [34].

Interestingly, the clients' and genetic counselors' experience in terms of the process in this study (Fig 1) was very similar to the process of genetic counseling itself, which indicates two possibilities. First, transcripts about empathic experiences may overlap with the process of genetic counseling. Sanders et al. [14], in a framework of medical relationships, reported categories of empathy similar to our results; care processes that helped patients feel better cared for were indicative of empathy. Furthermore, it was found that psychotherapists' empathy serves to facilitate and enhance the process of psychotherapy [12]. Considering previous studies, this might result in more information and experiences during the session being uncovered, because the process of genetic counseling was facilitated and enhanced, whereby the client was able to feel the counselor's empathy. Second, the process of empathy suggested by our study might show Japanese cultural empathy. Several studies have discussed the importance of culture and empathy [35–37]. The results of this study may only portray experiences in Japanese genetic counseling sessions. Existing literature shows that the Japanese recognize non-verbal cues in conversations and the mechanism of utterance interpretation relates to "*sasshi.*" "*Sasshi*" means "conjecture, surmise, or guessing what someone means" [38]. In its verb form (*sassuru*), its meaning is expanded to include imagining, empathizing with, and making allowances for others [39]. It is also reported that Japanese people attend more to the whole situation, including the context, than do North Americans [40]; people in high-contextual communication in East Asian cultures tend to have a holistic and dialectical mode of thought [41].

A culture of emphasizing nonverbal sensitivity to indirect form of communication might affect the empathic experience and its recall. This can be assessed further by simulated studies designed for cross-cultural research [42].

A limitation of this study is that all the participants in client roles were females in their 20s and Master's students, which might have resulted in the collection of largely similar experiences and perspectives. The client attributes assumed in the client scenarios were limited, to some extent, for the difficulties of using IPR for patients. However, our findings could be particularly applicable to clinical situations similar to our simulation settings, namely sessions with clients who have not been previously diagnosed or initial medical consultations. Counselors' behaviors during simulations may differ from usual behaviors; however, there is a report that simulations reproduced the variability of counseling behaviors including expressing empathy in surveys of clinical situations [43]. Moreover, a simulated genetic counseling consultation and a review using IPR may help inform more effective learning for practicing empathy. As our study used an online system, in-person communication was not explored and will be an aspect for further research to test our findings. There are also cultural and gender differences in communication characteristics [44, 45], and our data might only be representative of the perspectives of Japanese females.

## Conclusions

This is the first study to investigate a perceived empathy experience in simulated genetic counseling. This work lays the groundwork for future studies with real clients, as well as for cross-cultural studies in this area. The data obtained from the clients and genetic counselors suggested that clients' changes/lack of changes play a central role in the communication of empathy, which is a process whereby counselors try to address implicit aspects of their clients, and the clients are provided with time and a safe place for introspection. Our data suggested that the perception of empathy in a simulated genetic counseling consultation would be reported as a genetic counseling process that gave clients a better outlook, which also provides the opportunity for practitioners to learn skills related to empathy.

## Acknowledgments

This work was conducted to fulfill a degree requirement at Ochanomizu University. We would like to thank Professor Iwakabe Shigeru at Ritsumeikan University, formerly at Ochanomizu University for advising our IPR study. We appreciate the participants' willingness to be interviewed and their openness in sharing their personal information for the purposes of the study.

## Author Contributions

**Conceptualization:** Chikako Tomozawa, Motoko Sasaki, Hidehiko Miyake.

**Data curation:** Chikako Tomozawa, Mikiko Kaneko.

**Formal analysis:** Chikako Tomozawa, Mikiko Kaneko, Motoko Sasaki.

**Investigation:** Chikako Tomozawa, Motoko Sasaki.

**Methodology:** Chikako Tomozawa, Mikiko Kaneko, Hidehiko Miyake.

**Project administration:** Chikako Tomozawa.

**Resources:** Chikako Tomozawa.

**Supervision:** Hidehiko Miyake.

**Validation:** Chikako Tomozawa, Mikiko Kaneko, Motoko Sasaki, Hidehiko Miyake.

**Visualization:** Chikako Tomozawa.

**Writing – original draft:** Chikako Tomozawa.

**Writing – review & editing:** Mikiko Kaneko, Motoko Sasaki, Hidehiko Miyake.

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
