## [Decision Letter · Decision Letter 0]

31 May 2023

PONE-D-23-04985Clients’ and genetic counselors’ perceptions of empathy in Japan: A pilot study of simulated genetic counseling sessionsPLOS ONE

Dear Dr. Tomozawa,

Thank you for submitting your manuscript to PLOS ONE. After careful consideration, we feel that it has merit but does not fully meet PLOS ONE’s publication criteria as it currently stands. Therefore, we invite you to submit a revised version of the manuscript that addresses the points raised during the review process.

We look forward to receiving your revised manuscript.

Kind regards,

Ramona Bongelli, Ph.D.

Academic Editor

PLOS ONE

Journal Requirements:

Reviewers' comments:

Reviewer's Responses to Questions

**Comments to the Author**

1. Is the manuscript technically sound, and do the data support the conclusions?

Reviewer #1: Partly

Reviewer #2: Yes

2. Has the statistical analysis been performed appropriately and rigorously? 

Reviewer #1: N/A

Reviewer #2: N/A

3. Have the authors made all data underlying the findings in their manuscript fully available?

Reviewer #1: No

Reviewer #2: No

4. Is the manuscript presented in an intelligible fashion and written in standard English?

Reviewer #1: No

Reviewer #2: Yes

5. Review Comments to the Author

Reviewer #1: The English should be reviewed by a native speaker for clarity and meaning.

I will point out some of the specific sentences that need attention with “clarity/meaning:” when I dont know how to edit for clarity. And in other cases I suggest alternate wording.

Abstract:

Clarity/meaning: “Client-counselor interactions about empathy involve a significant element in genetic counseling, which is an area of increasing need with the rapid acceleration of genetic testing across medical conditions and needs.”

Suggest changing: “This study aims to reveal the perceptions of empathy in clients and genetic counselors during simulated genetic counseling sessions” to “This study aimed to explore the perceptions of empathy in simulated genetic counseling sessions from the perspectives of clients and genetic counselors”

Suggest changing: “Semi-structured interviews with participants of simulated genetic counseling sessions and interpersonal process recall were used to elicit their experiences of empathy” to: “Semi-structured interviews and interpersonal process recall were used with participants of simulated genetic counseling sessions to elicit their experiences of empathy”

Suggest changing: “A constructing grounded theory was used for data analysis” to: “A constructivist grounded theory was used for data analysis”

Suggest changing: “A total of 15 participants, 10 acting as clients and 5 as genetic counselors, participated in 10 simulated counseling sessions” to: “A total of 15 participants, including 10 clients and 5 genetic counselors, participated in 10 simulated counseling sessions.”

Clarity/meaning: “Clients’ perception represented genetic counselors’ empathic approaches, their subjective thoughts brought by the approaches and those changes”

Then suggest putting the sentence about counsellors attempts to demonstrate empathy first, and follow it with the sentence about clients reactions to the demonstrations of empathy.

Introduction

Suggest deleting “the psychometrics of” from this sentence: “Regarding empathy in the physician-patient relationship, the psychometrics of the patient’s view of the physician’s…” and changing the word “compliance” to “adherence”

Instead of “cancer patients” use “patients with cancer”

Suggest deleting “the sense of” in this sentence: “there is a lack of empirical data about the sense of empathy in genetic counselors.”

Change: “. The aim of this pilot study is to reveal the clients’ perceptions of empathy in the genetic counseling context by analyzing simulated sessions.” To ”. The aim of this pilot study was to explore counsellors’ and clients’ perceptions of empathy in the genetic counseling context by analyzing simulated sessions.”

[amemdment: having read the whole manuscript, it seems to be about THE PROCESS through which empathy is experienced in genetic counseling, rather than just empathy in general - suggest rewording the above accordingly]

Change: “However, IPR was difficult to adopt for clients in clinical settings for the feasibility of this study that was conducted to fulfill a degree requirement. Therefore, our survey targeted simulated genetic counseling session” to “Due to the difficulties associated with using IPR for clients in clinical settings, our survey targeted simulated genetic counseling session…”

Methods:

More information is required about the methodology chosen. Specifically why was grounded theory chosen ? How was it actually operationalized within a constructivist paradigm? This sentence makes it sounds actually rather like the approach taken was closer to positivist or post positivist than constructivist: “Coding and theorizing procedures were repeated to reach consensus between both author”. Suggest the authors examine https://onlinelibrary.wiley.com/doi/10.1002/jgc4.1644 Further, what was the underlying conceptual framework?

The following suggests that the authors are alluding to the concept of “saturation”: “Data collection was completed when new categories were not generated after new data addition” - however, the specifics and evidence of having reached saturation is missing.

Results

The data provided are very thin there are very few direct quotes from participants- the value of qualitative approaches is typically within the richness of the data, the depth of understanding of participants perspectives and experiences. In this manuscript, the few quotes that are provided are all within table 4, and are separated into counselor and client quotes. This presentation feels very positivist and does not allow for true appreciation of the way in which the different components of the model interrelate.

The authors indicate that they built a theoretical model - it’s not clear what exactly the model is of (perhaps see the comment above about the process of developing , and there is no overarching description of how all of the components of the model fit together.

Reviewer #2: Thank you for the opportunity to review this study exploring Japanese genetic counsellors’ experiences of empathy in simulated consultations. The study contributes to our understanding of what occurs between client and genetic counsellor within a clinical interaction. Use of simulated consultations and IPR builds on earlier work conducted in North America. The authors of this paper bring a reflective and considered Japanese cultural understanding to this process study.

The majority of the paper is very well written and easy to read. The abstract is not as clear as the rest of the paper and includes some grammatical errors. Editing the abstract to ensure the meaning is clear will help increase the profile of this work.

The term simulation can be used to mean a simulated consultation (as in this paper) or a learning experience (as described in a recent genetic counselling paper by Jacobs & McEwen, 2021). Simulation can also be used as a form of assessment. It would be helpful to include a definition of simulation as used by these authors, and to consider whether simulated consultation is a more appropriate term.

The authors discuss the possibility of using simulation to develop empathy in genetic counselling education and training. Including literature where simulation has been used with learning outcomes that include developing effective client-counsellor relationships would further support the discussion. This paper may be of interest.

Jacobs, C. and McEwen, A. (2021). Adapting to the challenges of the global pandemic on genetic counselor education: Evaluating students’ satisfaction with virtual clinical experiences. J Genet Couns. 2021;30:1074–1083 DOI: 10.1002/jgc4.1490

I encourage the authors to proofread the manuscript prior to publication to correct any minor errors.

Thank you for this work, demonstrating the value of simulated consultations in understanding what occurs within a genetic counselling consultation.

6. PLOS authors have the option to publish the peer review history of their article (what does this mean?). If published, this will include your full peer review and any attached files.

Reviewer #1: No

Reviewer #2: No

---

## [Author Response · Author response to Decision Letter 0]

13 Jun 2023

Dr. Ramona Bongelli

Academic Editor

PLOS ONE

Dear Editor

Re: Manuscript ID: PONE-D-23-04985

Dear Dr. Bongelli,

 Thank you very much for your e-mail and review of the manuscript (PONE-D-23-04985) that we sent on February 20, 2023. We thank both the reviewers for providing constructive comments regarding the improvement of our original manuscript. 

Here, we are sending our revised manuscript that has benefitted from your insightful suggestions. All changes have been made in response to the reviewers’ comments, and itemized responses to each reviewer’s comments are also attached. 

The authors received no specific funding for this work. The data that support the findings of this study are available on request from the Institutional Review Board of the Humanities and Social Science Research Ethics Committee in Ochanomizu University (e-mail: kenkyo-TL@cc.ocha.ac.jp) and the corresponding author. The data are not publicly available due to privacy and ethical restrictions. Parts of data that support the findings of this study are available in this article.

Reviewer #1:

Q1. The English should be reviewed by a native speaker for clarity and meaning.

A1. Thank you for your comment. The paper has been edited and rewritten by an experienced scientific editor again, who has improved the grammar and stylistic expression of the manuscript.

Abstract:

Q2. Clarity/meaning: “Client-counselor interactions about empathy involve a significant element in genetic counseling, which is an area of increasing need with the rapid acceleration of genetic testing across medical conditions and needs.”

A2. This sentence has been corrected to “Client-counselor interactions about empathy involve a significant element in genetic counseling, which is an area of increasing need with the rapid acceleration of genetic testing.”

Q3. Suggest changing: “This study aims to reveal the perceptions of empathy in clients and genetic counselors during simulated genetic counseling sessions” to “This study aimed to explore the perceptions of empathy in simulated genetic counseling consultations from the perspectives of clients and genetic counselors.”

A3. As indicated, the sentence has been changed. The term “consultations” has been used instead of “sessions” to refer to Reviewer 2’s comment. Kindly refer to Reviewer 2’s comments for your reference.

Q4. Suggest changing: “Semi-structured interviews with participants of simulated genetic counseling sessions and interpersonal process recall were used to elicit their experiences of empathy” to: “Semi-structured interviews and interpersonal process recall were used with participants of simulated genetic counseling consultations to elicit their experiences of empathy.”

A4. As indicated, the sentence has been changed.

Q5. Suggest changing: “A constructing grounded theory was used for data analysis” to: “A constructivist grounded theory was used for data analysis.”

A5. As indicated, the wording has been changed.

Q6. Suggest changing: “A total of 15 participants, 10 acting as clients and 5 as genetic counselors, participated in 10 simulated counseling sessions” to: “A total of 15 participants, including 10 clients and 5 genetic counselors, participated in 10 simulated counseling consultations.”

A6. As indicated, the sentence has been changed.

Q7. Clarity/meaning: “Clients’ perception represented genetic counselors’ empathic approaches, their subjective thoughts brought by the approaches and those changes”

Then suggest putting the sentence about counsellors attempts to demonstrate empathy first, and follow it with the sentence about clients reactions to the demonstrations of empathy.

A7. Thank you for your valuable suggestion. As per your recommendation, these sentences have been changed to “The genetic counselors attempted to demonstrate empathy and were sensitive toward detecting changes in clients. Meanwhile, the clients’ perceptions represented their feelings and thoughts elicited through the counselors’ empathic approaches.”

Introduction

Q8. Suggest deleting “the psychometrics of” from this sentence: “Regarding empathy in the physician-patient relationship, the psychometrics of the patient’s view of the physician’s…” and changing the word “compliance” to “adherence”

A8. As indicated, the wording has been changed.

Q9. Instead of “cancer patients” use “patients with cancer”

A9. As indicated, the wording has been changed.

Q10. Suggest deleting “the sense of” in this sentence: “there is a lack of empirical data about the sense of empathy in genetic counselors.”

A10. As indicated, the sentence has been changed.

Q11. Change: “. The aim of this pilot study is to reveal the clients’ perceptions of empathy in the genetic counseling context by analyzing simulated sessions.” To ”. The aim of this pilot study was to explore counsellors’ and clients’ perceptions of empathy in the genetic counseling context by analyzing simulated sessions.”

[amemdment: having read the whole manuscript, it seems to be about THE PROCESS through which empathy is experienced in genetic counseling, rather than just empathy in general - suggest rewording the above accordingly]

A11. Thank you for your valuable feedback. This sentence has been corrected to “ The aim of this pilot study was to explore the process through which counselors and clients experienced empathy in the genetic counseling context by analyzing simulated consultations.”

Q12. Change: “However, IPR was difficult to adopt for clients in clinical settings for the feasibility of this study that was conducted to fulfill a degree requirement. Therefore, our survey targeted simulated genetic counseling session” to “Due to the difficulties associated with using IPR for clients in clinical settings, our survey targeted simulated genetic counseling session…”

A12. Thank you for your suggestion. As indicated, the sentence has been changed, and the term “consultations” has been used instead of “sessions.”

Methods

Q13. More information is required about the methodology chosen. Specifically why was grounded theory chosen ? How was it actually operationalized within a constructivist paradigm? This sentence makes it sounds actually rather like the approach taken was closer to positivist or post positivist than constructivist: “Coding and theorizing procedures were repeated to reach consensus between both author”. Suggest the authors examine https://onlinelibrary.wiley.com/doi/10.1002/jgc4.1644 Further, what was the underlying conceptual framework?

A13. Thank you for your comment. As per your suggestions, the Data analysis section of our Methods has been thoroughly rephrased.

Q14. The following suggests that the authors are alluding to the concept of “saturation”: “Data collection was completed when new categories were not generated after new data addition” - however, the specifics and evidence of having reached saturation is missing.

A14. Thank you for your comment. The concept of “theoretical sufficiency” and its specifics have been added in the Data analysis section.

Results

Q15. The data provided are very thin there are very few direct quotes from participants. the value of qualitative approaches is typically within the richness of the data, the depth of understanding of participants perspectives and experiences. In this manuscript, the few quotes that are provided are all within table 4, and are separated into counselor and client quotes. This presentation feels very positivist and does not allow for true appreciation of the way in which the different components of the model interrelate.

The authors indicate that they built a theoretical model - it’s not clear what exactly the model is of (perhaps see the comment above about the process of developing , and there is no overarching description of how all of the components of the model fit together.

A15. Thank you for your valuable comments. The Results and Discussion sections have been thoroughly rephrased.

Reviewer #2: 

Q1. The abstract is not as clear as the rest of the paper and includes some grammatical errors. Editing the abstract to ensure the meaning is clear will help increase the profile of this work.

A1. Thank you for your comment. The abstract has been revised to clarify meanings and correct any grammatical errors.

Q2. The term simulation can be used to mean a simulated consultation (as in this paper) or a learning experience (as described in a recent genetic counselling paper by Jacobs & McEwen, 2021). Simulation can also be used as a form of assessment. It would be helpful to include a definition of simulation as used by these authors, and to consider whether simulated consultation is a more appropriate term.

A2. Thank you for your comment. We have changed our use of the term “simulation” to “simulated consultation” at all relevant instances in the manuscript.

Q3. The authors discuss the possibility of using simulation to develop empathy in genetic counselling education and training. Including literature where simulation has been used with learning outcomes that include developing effective client-counsellor relationships would further support the discussion. This paper may be of interest.

Jacobs, C. and McEwen, A. (2021). Adapting to the challenges of the global pandemic on genetic counselor education: Evaluating students’ satisfaction with virtual clinical experiences. J Genet Couns. 2021;30:1074–1083 DOI: 10.1002/jgc4.1490

A3. Thank you for your helpful suggestion. We have added the paper (Jacobs & McEwen 2021) in the reference list.

---

## [Decision Letter · Decision Letter 1]

29 Jun 2023

PONE-D-23-04985R1Clients’ and genetic counselors’ perceptions of empathy in Japan: A pilot study of simulated consultations of genetic counselingPLOS ONE

Dear Dr. Tomozawa,

Thank you for submitting your manuscript to PLOS ONE. After careful consideration, we feel that it has merit but does not fully meet PLOS ONE’s publication criteria as it currently stands. Therefore, we invite you to submit a revised version of the manuscript that addresses the points raised during the review process.==============================

Dear authors,

thank you first of all for submitting such an interesting article. 

In line with what one of the reviewers suggested, I would kindly ask you to make the suggested changes. 

We look forward to receiving your revised manuscript.

Kind regards,

Ramona Bongelli, Ph.D.

Academic Editor

PLOS ONE

Journal Requirements:

Additional Editor Comments:

Dear Authors,

Thank you for reviewing your paper.

I would ask you to make the last minor revisions suggested by one of the reviewers.

Good work

Ramona Bongelli

Reviewers' comments:

Reviewer's Responses to Questions

**Comments to the Author**

1. If the authors have adequately addressed your comments raised in a previous round of review and you feel that this manuscript is now acceptable for publication, you may indicate that here to bypass the “Comments to the Author” section, enter your conflict of interest statement in the “Confidential to Editor” section, and submit your "Accept" recommendation.

Reviewer #1: (No Response)

2. Is the manuscript technically sound, and do the data support the conclusions?

Reviewer #1: Yes

3. Has the statistical analysis been performed appropriately and rigorously? 

Reviewer #1: N/A

4. Have the authors made all data underlying the findings in their manuscript fully available?

Reviewer #1: Yes

5. Is the manuscript presented in an intelligible fashion and written in standard English?

Reviewer #1: Yes

6. Review Comments to the Author

Reviewer #1: This manuscript has improved markedly from the first submission, the authors are commended for their work.

The following comments are offered to further improve the usefulness of the manuscript.

Abstract:

Suggest changing the first two sentences of the abstract to read as follows:

The rapid increase of the availability of genetic testing is driving the acceleration of genetic counseling implementation. Empathy is important in medical encounters in general, and forms a core component of a successful genetic counseling session, but there ilittle empirical data about empathy in genetic counseling.

Methods

The authors discuss that they are male and female, and expect that the participants would be female… but its not actually clear in how the language is being used whether the authors are referring to sex or gender. Clarity on this would be useful. Especially as the discussion comments: “There are also cultural and gender differences in communication characteristics [43,44], and our data might only be representative of the perspectives of Japanese women.” Please note that while “women” is a term that refers only to gender and not sex, “female” is typically used to refer to sex but can also be used to refer to gender.

It would also be useful to make explicit whether counsellor and client IPR interviews were conducted separately or together.

How was it decided which sections of the video related to empathy, and who made that decision? This should be included in the manuscript.

The rationale for choosing grounded theory as a methodology is not strong as currently written. Specifically, there are many approaches that can apply the constant comparative method ( described in lines 157-132), rather not all qualitative methodologies aim to produce a theory of a process - this seems to be the most compelling reason to choose this methodology for this work. Suggest rewording accordingly.

Please check through the manuscript and ensure that themes are not referred to as “emerging” (e.g. p31)n- as this term suggests that they are objectively present, which runs counter to the premise of constructivism

Discussion

having participated as a counsellor in a simulated genetic counseling session, I wonder whether there is any chance that the fact that it is simulated might interfere with or influence the ways in which empathy manifests in a session. This might warrant mention in the discussion (if there is any literature about this) or limitations.

7. PLOS authors have the option to publish the peer review history of their article (what does this mean?). If published, this will include your full peer review and any attached files.

Reviewer #1: No

---

## [Author Response · Author response to Decision Letter 1]

4 Jul 2023

Reviewer #1:

Abstract:

Q1. Suggest changing the first two sentences of the abstract to read as follows:

The rapid increase of the availability of genetic testing is driving the acceleration of genetic counseling implementation. Empathy is important in medical encounters in general, and forms a core component of a successful genetic counseling session, but there ilittle empirical data about empathy in genetic counseling.

A1. As recommended, the first two sentences have been revised.

Methods

Q2. The authors discuss that they are male and female, and expect that the participants would be female… but its not actually clear in how the language is being used whether the authors are referring to sex or gender. Clarity on this would be useful. Especially as the discussion comments: “There are also cultural and gender differences in communication characteristics [43,44], and our data might only be representative of the perspectives of Japanese women.” Please note that while “women” is a term that refers only to gender and not sex, “female” is typically used to refer to sex but can also be used to refer to gender.

A2. Thank you for your valuable suggestion. As per your recommendation, we have revised the sentence under the section on Researchers to clarify the cisgender identity of the research team members. Further, the phrase “women” has been changed to “females.”

Q3. It would also be useful to make explicit whether counsellor and client IPR interviews were conducted separately or together.

A3. Thank you for your valuable suggestion. As per your recommendation, the fact that counsellor and client IPR interviews were conducted individually has been added.

Q4. How was it decided which sections of the video related to empathy, and who made that decision? This should be included in the manuscript.

A4. As indicated, the sentence has been revised to highlight that each interviewee decided the sections of the video related to empathy.

Q5. The rationale for choosing grounded theory as a methodology is not strong as currently written. Specifically, there are many approaches that can apply the constant comparative method ( described in lines 157-132), rather not all qualitative methodologies aim to produce a theory of a process - this seems to be the most compelling reason to choose this methodology for this work. Suggest rewording accordingly.

A5. As indicated, the rationale for choosing the method has been revised to highlight the suitability of grounded theory for the purpose of producing a theory.

Q6. Please check through the manuscript and ensure that themes are not referred to as “emerging” (e.g. p31)n- as this term suggests that they are objectively present, which runs counter to the premise of constructivism.

A6. As per your recommendation, the word “emerged” has been changed to “were generated.” 

Discussion

Q7. Having participated as a counsellor in a simulated genetic counseling session, I wonder whether there is any chance that the fact that it is simulated might interfere with or influence the ways in which empathy manifests in a session. This might warrant mention in the discussion (if there is any literature about this) or limitations.

A7. Thank you for your valuable comments. We searched the literature for evidence regarding whether a simulated situation influenced the ways in which empathy manifests; however, no clear evidence was found. We have mentioned the possibility of such an influence in the Discussion and have added the following reference regarding practitioners’ behaviors in simulations:

[43] Boss RD, Donohue PK, Roter DL, Larson SM, Arnold RM. “This is a decision you have to make”: using simulation to study prenatal counseling. Simul Healthc. 2012;7: 207–212. doi: 10.1097/SIH.0b013e318256666a

---

## [Editor Report · Decision Letter 2]

6 Jul 2023

Clients’ and genetic counselors’ perceptions of empathy in Japan: A pilot study of simulated consultations of genetic counseling

PONE-D-23-04985R2

Dear Dr. Tomozawa,

We’re pleased to inform you that your manuscript has been judged scientifically suitable for publication and will be formally accepted for publication once it meets all outstanding technical requirements.

Kind regards,

Ramona Bongelli, Ph.D.

Academic Editor

PLOS ONE
---

## [Editor Report · Acceptance letter]

10 Jul 2023

PONE-D-23-04985R2 

Clients’ and genetic counselors’ perceptions of empathy in Japan: A pilot study of simulated consultations of genetic counseling 

Dear Dr. Tomozawa:

I'm pleased to inform you that your manuscript has been deemed suitable for publication in PLOS ONE. Congratulations! Your manuscript is now with our production department. 

Kind regards, 

on behalf of

Professor Ramona Bongelli 

Academic Editor

PLOS ONE